# SKILL: Structural Knowledge Injection into Large Language Models for Inductive Knowledge Graph Reasoning

## Abstract

Knowledge Graph Reasoning (KGR) aims to predict missing (head, relation, tail) triples by inferring new facts from existing ones within a knowledge graph. While recent methods embed entities and relations into vectors or model multi-hop paths, they predominantly rely on statistical co-occurrence patterns, yielding logically inconsistent or semantically implausible paths that degrade prediction quality. We introduce SKILL, a new framework that revolutionizes KGR by injecting structural knowledge into large language models (LLMs) through inductive reasoning, thereby optimizing the reasoning process with LLMs' semantic understanding capabilities. Our novel rule-miner module extracts and semantically validates symbolic reasoning rules from closed paths using LLM-based one-shot prompting, effectively filtering out invalid patterns. This innovative rule injection fine-tunes LLMs with explicit symbolic guidance, leading to a comprehension of KG structures required for downstream reasoning. Extensive experiments on three standard inductive benchmarks show that SKILL surpasses competing baselines by up to 5 absolute Hit@1 points, establishing a new state of the art for inductive knowledge graph reasoning.

## 1 Introduction

Knowledge Graphs (KGs) are structured representations of real-world entities and their relationships, typically modeled as directed graphs where nodes denote entities and edges correspond to relations. Each fact in a KG is represented as a triple $(h, r, t)$, indicating that a head entity $h$ is linked to a tail entity $t$ via a relation $r$. KGs have been widely applied in various downstream tasks, such as question answering Luo et al. (2024), dialogue systems Xu et al. (2019), and recommendation systems Guo et al. (2020). However, real-world KGs are often incomplete, motivating Knowledge Graph Reasoning (KGR), which seeks to infer missing facts from existing structures.

Traditional KGR approaches mainly rely on embedding models that encode entities and relations into low-dimensional vector spaces, learning scoring functions to assess the plausibility of candidate triples Bordes et al. (2013); Yang et al. (2015). More recent efforts have integrated path-based reasoning, modeling relational paths between entities as a structured source of relational knowledge Zhang et al. (2022); Cheng et al. (2023). Despite their notable progress, existing KGR methods face two fundamental limitations that hinder their effectiveness in real-world applications.

First, embedding-based models effectively capture local statistical patterns but often exhibit limited inductive generalization, especially in scenarios involving previously unseen entities Chen et al. (2023). Since these models operate in a latent embedding space, they offer little interpretability of the reasoning process and typically assume a closed-world setting. Consequently, they are unreliable in dynamic or incomplete KGs, as shown in Fig. 1. This limitation restricts their applicability in real-world settings, where new entities frequently emerge and explicit reasoning is often required.

Second, while path-based reasoning enhances structural awareness by leveraging relational paths, many existing approaches depend on co-occurrence heuristics to extract or rank them without verifying semantic validity Liang et al. (2024). As a result, they often propagate noisy or spurious paths that lack meaningful logical dependencies, thereby weakening the reliability of the inferred triples.

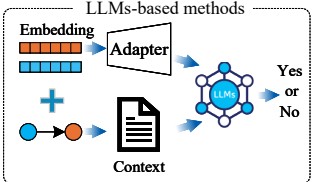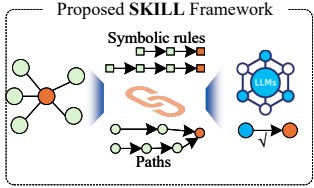

Figure 1: An illustration of current KGR methods. SKILL bridges KGs and LLMs with logical reasoning patterns.

Recent advances in LLMs have demonstrated impressive capabilities in understanding and generalizing from symbolic and natural language patterns. To incorporate structured information from Knowledge Graphs (KGs), recent works Zhang et al. (2024a;b); Guo et al. (2025) introduce adapter modules that map KG embeddings into the token representations of LLMs, enabling effective utilization of KG semantics. However, these methods usually rely on static KG embeddings, which are pre-trained independently of LLMs and often misaligned with contextualized token representations. Moreover, such approaches tend to treat the KG as an external memory rather than integrating its relational structure into the reasoning process, limiting the model's ability to perform relational and inductive reasoning over unseen entities or facts. Beyond adapter-based methods, recent works Pan et al. (2024); Luo et al. (2025) have explored using LLMs to directly mine reasoning rules from KGs, but often encounter issues of insufficient supervision and reduced robustness in noisy KG settings.

To address these challenges, we advocate a structural perspective on how KGs should support LLM-based reasoning. Real-world KGs typically involve numerous entities and complex relational structures, posing significant challenges for LLMs in accurately recognizing entities and relationships. Without explicit structural guidance, LLMs often fail to capture underlying logical patterns, limiting their inductive generalization ability. Instead of serving merely as static memory, KGs should provide symbolic relational patterns to explicitly guide LLMs toward structurally grounded reasoning. Such an approach requires extracting high-quality symbolic rules explicitly capturing the inherent logic of the KG. Injecting this structured knowledge into LLMs enhances interpretability and substantially improves inductive reasoning, particularly over unseen entities and sparse subgraphs.

Motivated by this structural perspective, we propose SKILL, a novel framework that explicitly injects structural knowledge from KGs into LLMs to enhance inductive reasoning capabilities. SKILL first identifies closed paths in the KG to derive candidate symbolic rules, then leverages one-shot prompting with an LLM to verify their semantic validity and filter out noisy or spurious patterns. The validated rules are then injected into LLMs via a logic-enhanced reasoning module, directly facilitating inductive generalization. By bridging symbolic reasoning and neural language modeling, SKILL enables robust and interpretable reasoning over KGs. Our contributions are threefold:

- We propose **SKILL**, a novel framework that explicitly injects structurally grounded symbolic rules from KGs into LLMs via one-shot prompting and logic-enhanced fine-tuning, significantly improving inductive reasoning.

- We introduce a logic-enhanced reasoning module that leverages semantically validated symbolic rules to explicitly guide LLM reasoning, substantially enhancing interpretability and inductive generalization.

- We advocate a structural perspective on KG-LLM integration, emphasizing the importance of high-quality symbolic rules in supporting inductive reasoning over unseen entities and sparse relational contexts.

Extensive experiments on three benchmarks—FB15k-237, WN18RR, and NELL-995—show that SKILL achieves comparable performance to state-of-the-art methods under transductive settings and significantly outperforms them under inductive scenarios. Notably, SKILL attains up to a 5% absolute improvement in Hit@1, demonstrating enhanced generalization capabilities over previously unseen entities. Furthermore, comprehensive ablation studies across multiple configurations confirm the effectiveness and necessity of each component in SKILL.

## 2 RELATED WORK

**Embedding-based KG Reasoning**: Embedding-based KG reasoning methods represent entities and relations as continuous vectors learned from observed triples, and score candidate facts using predefined functions to rank top-$k$ predictions, as in TransE Bordes et al. (2013), ComplEx Trouillon et al. (2016) and Adaprop Zhang et al. (2023). These methods are typically categorized into translational models Lin et al. (2015); Zhang et al. (2020), tensor decompositional models Balazevic et al. (2019); Zhang et al. (2019), and neural network models Dettmers et al. (2018); Zhang & Yao (2022). Despite their progress, these methods still lack interpretability and generalize poorly to inductive scenarios with unseen entities Chen et al. (2020). Moreover, because reasoning occurs entirely in latent spaces, inferred triples cannot be traced back to human-understandable logic.

**Path-based KG Reasoning**: Path-based methods capture logical dependencies between head and tail entities by exploring multi-hop relational paths in the KG. The Path Ranking Algorithm (PRA) Lao & Cohen (2010) applies path-constrained random walks to mine relational rules, forming the basis for later path-based reasoning. Subsequent approaches exploit the compositional semantics of relation chains to enable more structured and explainable reasoning Yang et al. (2017); Cheng et al. (2022; 2023). However, many still rely on surface-level co-occurrence or heuristic sampling, which introduces semantically invalid or noisy paths Liang et al. (2024). This reliance on statistical correlations rather than logical validity undermines robustness, particularly in sparse or inductive settings where meaningful paths are scarce.

**LLM-based KG Reasoning**: LLM-based methods leverage the strong contextual understanding of large language models for KGR. A common approach reformulates triples or multi-hop relational paths into natural language sequences, allowing LLMs to perform reasoning tasks via prompting Yao et al. (2019); Su et al. (2023; 2024); Wu et al. (2024). Such methods exploit the pre-trained knowledge of LLMs and reduce reliance on task-specific training, but they lack structured inductive bias and generalize poorly to unseen entities or relations Pan et al. (2024). To address these limitations, recent efforts have proposed adapter-based integration strategies Zhang et al. (2024b); Jiang et al. (2024); Zhang et al. (2024a); Guo et al. (2025). These approaches typically encode entity and relation embeddings separately and inject them into LLMs as additional token embeddings or prefix prompts, aligning symbolic KG information with contextual representations.

In contrast to prior works that either inject static embeddings or rely on heuristic rule induction, our work takes a structural perspective by treating KGs as sources of relational logic. We introduce a logic-enhanced reasoning module that enables LLMs to reason over high-quality, semantically validated relational rules, thereby enhancing inductive reasoning and interpretability.

## 3 PRELIMINARIES

**Knowledge Graphs** Let $\mathcal{G} = (\mathcal{E}, \mathcal{R}, \mathcal{T})$ denote a knowledge graph, where $\mathcal{E}$ is the set of entities, $\mathcal{R}$ is the set of relation types, and $\mathcal{T} \subseteq \mathcal{E} \times \mathcal{R} \times \mathcal{E}$ is the set of factual triples. Each triple $(h, r, t) \in \mathcal{T}$ indicates that the head entity $h$ is connected to the tail entity $t$ via relation $r$.

**Knowledge Graph Reasoning** Knowledge graph reasoning (KGR) aims to infer missing facts from the existing triples in $\mathcal{G}$ by capturing and generalizing relational patterns among entities. Specifically, given a query in the form of an incomplete triple $(h, r, ?)$ or $(?, r, t)$, the task is to predict the most plausible tail or head entity, respectively. In the inductive setting, the entity sets in the training and test knowledge graphs are disjoint, i.e., $\mathcal{E}_{\text{train}} \cap \mathcal{E}_{\text{test}} = \emptyset$, $\mathcal{R}_{\text{test}} \subseteq \mathcal{R}_{\text{train}}$.

Inductive reasoning poses unique challenges, as the model must reason over unseen entities without direct exposure during training. Unlike the transductive setting, where all entities are present during training and representations can be learned directly, inductive reasoning requires the model to generalize based on entity attributes, relation semantics, and local graph structure. This becomes particularly difficult in cases with sparse textual descriptions, limited neighborhood information, or highly heterogeneous relations.

**Symbolic Reasoning Rules** Symbolic reasoning rules are a sequence of relations $(r_1, r_2, \ldots, r_k)$ connecting a head entity $h$ to a tail entity $t$ through a sequence of intermediate entities $(e_1, e_2, \ldots, e_{k-1})$, forming a path such as $h \xrightarrow{r_1} e_1 \xrightarrow{r_2} \cdots \xrightarrow{r_k} t$.

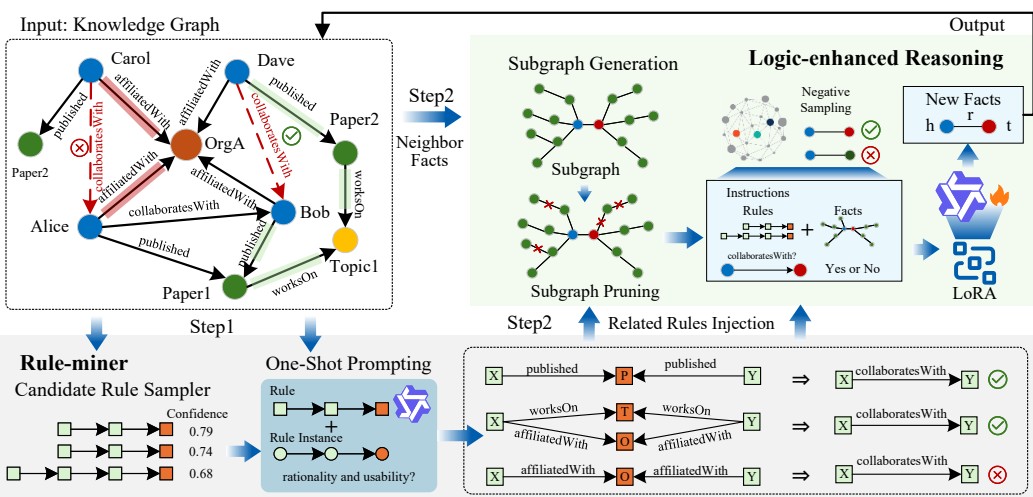

Figure 2: The overview of SKILL framework for inductive knowledge graph reasoning.

Recent approaches attempt to mine such symbolic reasoning rules from multi-hop paths in the graph, aiming to provide interpretable and generalizable reasoning patterns. However, existing methods often rely on shallow co-occurrence statistics or frequent path patterns, which may lead to the extraction of semantically invalid or spurious rules. These unreliable paths degrade both the interpretability and reasoning quality of predicted triples when applied to inductive scenarios. In particular, models struggle to distinguish between meaningful logical dependencies and noisy correlations, especially when applied to previously unseen entities or sparse subgraphs.

## 4 METHOD

Figure 2 illustrates the overall architecture of SKILL, a framework that transfers structural knowledge of KGs into LLMs via deductive reasoning. The framework consists of two major components: (1) a rule-miner module that automatically discovers, evaluates, and filters symbolic reasoning rules from a given KG; and (2) a logic-enhanced reasoning module that encodes these rules into the LLM via instruction-style fine-tuning, enabling the model to perform generalizable and interpretable reasoning on unseen entities and relational patterns.

### 4.1 RULE-MINER

Although substantial progress has been made in mining symbolic rules from KGs to enhance interpretability in reasoning tasks, existing approaches often lack semantic grounding. Most rule-mining methods prioritize statistical signals—such as path frequency or confidence scores—while neglecting whether the extracted rules are semantically coherent or aligned with commonsense or domain knowledge. As a result, rules that are syntactically valid may still be semantically implausible, introducing noise in downstream tasks.

For instance, a strong statistical association might exist between a person's gender and their marital status, yet it is semantically incorrect to infer one from the other—gender does not determine whether someone is married, and vice versa. Such spurious correlations can lead to misleading or biased reasoning if not properly filtered.

To address this issue, the rule-miner module adopts a two-stage validation strategy. In the first stage, it performs reasoning path sampling by enumerating closed paths of a predefined length $k$. To effectively capture the structural knowledge, we adopt a breadth-first search (BFS) based sampler to systematically generate these closed paths as candidate instances for reasoning rules. These closed paths serve as the symbolic patterns on which the subsequent rule mining and validation processes

operate. Given a triple $(h, r, t)$, define the set of length-$k$ supporting paths from $h$ to $t$ as:

$$\mathcal{P}_k(h, t) = \left\{ h \xrightarrow{r_1} e_1 \xrightarrow{r_2} \cdots \xrightarrow{r_k} t \; : \; (e_{i-1}, r_i, e_i) \in \mathcal{T} \; \forall i \right\}. \tag{1}$$

A path $\pi \in \mathcal{P}_k(h, t)$ is called *closed w.r.t.* $(h, r, t)$ if $(h, r, t) \in \mathcal{T}$. Then, we can extract a reasoning rule of the form:

$$\rho : r_1(x, e_1) \wedge r_2(e_1, e_2) \wedge \cdots \wedge r_k(e_{k-1}, y) \Rightarrow r(x, y), \tag{2}$$

where $x = h$, $y = t$, and $r$ is the target relation to be predicted. This rule expresses that if the body (left-hand side) relations hold along the path from $x$ to $y$, then it is likely that the relation $r(x, y)$ also holds. Such rules encode multi-hop structural dependencies and provide interpretable, compositional reasoning patterns for downstream tasks.

To quantify the reliability of a candidate rule $\rho$, we compute its *support* and *confidence*. Let $\mathrm{Body}_\rho(x, y)$ denote that there exist $e_1, \ldots, e_{k-1}$ such that $(x, r_1, e_1), \ldots, (e_{k-1}, r_k, y) \in \mathcal{T}$. The **support** of $\rho$ is the number of entity pairs $(x, y)$ for which both the body and the head triple $(x, r, y)$ appear in the KG:

$$\mathrm{supp}(\rho) = \left| \{(x, y) \; : \; \mathrm{Body}_\rho(x, y) \; \wedge \; (x, r, y) \in \mathcal{T}\} \right|. \tag{3}$$

The **confidence** of $\rho$ is the conditional probability that the head holds given that the body is satisfied:

$$\mathrm{conf}(\rho) = \frac{\mathrm{supp}(\rho)}{\left| \{(x, y) \; : \; \mathrm{Body}_\rho(x, y)\} \right|}, \tag{4}$$

with the convention that $\mathrm{conf}(\rho) = 0$ if the denominator is zero.

In the second stage, we further assess the semantic validity of rules via LLM-based one-shot prompting. While support and confidence capture empirical regularities, they cannot determine whether a rule is semantically plausible or aligned with commonsense.

We leverage LLMs as external semantic priors: a rule-to-text translation maps $\rho$ to a natural-language statement, which is then instantiated with a concrete entity pair $(x^*, y^*)$ satisfying $\mathrm{Body}_\rho(x^*, y^*)$. The LLM receives a one-shot prompt and returns a binary plausibility judgment:

$$\mathrm{LLM\_valid}(\rho) = \begin{cases} 1, & \text{if the model answers ``Yes'',} \\ 0, & \text{if the model answers ``No''.} \end{cases} \tag{5}$$

We retain only rules with $\mathrm{LLM\_valid}(\rho) = 1$, thereby filtering out semantically implausible or spurious patterns.

## 4.2 Logic-Enhanced Reasoning

After obtaining a set of semantically validated reasoning rules, the logic-enhanced reasoning module aims to transfer this structured knowledge into a large language model (LLM) to enhance its inductive reasoning capabilities. Specifically, we incorporate the filtered rules through instruction-based fine-tuning, enabling the model to learn to reason over unseen entities and sparse relational contexts by leveraging interpretable patterns.

### 4.2.1 Reasoning Subgraph Generation

Given a query triple $(h, r, t)$, we construct a reasoning subgraph $\mathcal{G}_{(h, r, t)}$ to provide structural context for assessing its plausibility. The subgraph is composed of two components as below.

**First-order neighborhood**: For each entity $e \in \{h, t\}$, we retrieve all directly connected triples:

$$\mathcal{N}_1(e) = \{(e, r', e') \in \mathcal{T}\} \cup \{(e', r', e) \in \mathcal{T}\}. \tag{6}$$

**Closed paths**: We extract all relational paths $\pi = (r_1, r_2, \ldots, r_l)$ of length $l \leq k$ that form a connection from $h$ to $t$ via intermediate entities, i.e., $\pi : h \xrightarrow{r_1} e_1 \xrightarrow{r_2} \cdots \xrightarrow{r_l} t$, and include all triples involved in such paths. We define the resulting reasoning subgraph as:

$$\mathcal{G}_{(h, r, t)} = \mathcal{N}_1(h) \cup \mathcal{N}_1(t) \cup \bigcup_{\pi \in \mathcal{P}_{h \to t}} \mathrm{Triples}(\pi), \tag{7}$$

where $\mathcal{P}_{h \to t}$ denotes the set of all closed paths from $h$ to $t$ of length at most $k$, and $\mathrm{Triples}(\pi)$ denotes the triples along each closed path.

### 4.2.2 REASONING SUBGRAPH PRUNING

The reasoning subgraph $\mathcal{G}_{(h,r,t)}$ may contain numerous closed paths connecting $h$ and $t$. However, not all paths fully satisfy the body of any validated rule, and strict filtering might discard useful partial evidence. To address this issue, we employ a soft matching strategy combined with confidence-weighted filtering to select relevant rules adaptively.

For each closed path $\pi$ and candidate rule $\rho$, we define a matching score $\text{match}(\pi, \rho) \in [0, 1]$ measuring the fraction of the rule's body premises covered by the path. If the body of $\rho$ contains $n$ relation atoms, $\rho : r_1(x, e_1) \wedge r_2(e_1, e_2) \wedge \cdots \wedge r_n(e_{n-1}, y) \Rightarrow r(x, y)$, and $\pi$ covers $k \leq n$ of these relations in order, then

$$\text{match}(\pi, \rho) = \frac{k}{n}. \tag{8}$$

Each rule $\rho$ has an associated confidence score $\text{conf}(\rho) \in [0, 1]$ reflecting its reliability. We compute a combined relevance score for each path-rule pair:

$$s(\pi, \rho) = \text{match}(\pi, \rho) \cdot \text{conf}(\rho). \tag{9}$$

We retain the top-$K$ path-rule pairs with the highest scores:

$$\mathcal{S}_{(h,r,t)} = \text{TopK}(\{(\pi, \rho)\}, \ s(\pi, \rho), \ K), \tag{10}$$

where the candidate pairs $(\pi, \rho)$ are drawn from the closed path set $\mathcal{P}(h, t)$ and the validated rule set $R$ with head $r$, subject to the condition $\text{match}(\pi, \rho) > 0$. Then, the pruned reasoning subgraph is defined as

$$\tilde{\mathcal{G}}_{(h,r,t)} = \bigcup_{(\pi, \rho) \in \mathcal{S}_{(h,r,t)}} \text{Triples}(\pi), \tag{11}$$

containing only triples on the most relevant paths. Correspondingly, the relevant rule set for reasoning is adaptively determined by

$$R_{(h,r,t)} = \{\rho \mid \exists \pi \text{ with } (\pi, \rho) \in \mathcal{S}_{(h,r,t)}\}. \tag{12}$$

We select neighboring triples that align with candidate rules and exhibit high semantic similarity to the query triple. Specifically, each triple $(h, r, t)$ and its neighbor $(h', r', t')$ are first converted into natural language text $T(h, r, t)$ and $T(h', r', t')$. These textual representations are then encoded into embeddings with a pre-trained encoder $f(\cdot)$ Chen et al. (2024), which provides semantic representations that preserve both structural alignment and contextual coherence. The similarity is computed via cosine similarity:

$$\text{sim}(T(h, r, t), \ T(h', r', t')) \ = \ \frac{f(T(h, r, t)) \ \cdot \ f(T(h', r', t'))}{\|f(T(h, r, t))\| \ \|f(T(h', r', t'))\|}. \tag{13}$$

The fine-tuning objective minimizes the cross-entropy loss over a dataset of prompts and binary labels. Given a training pair $(P, y)$, where $P$ is the prompt and $y \in \{0, 1\}$ is the ground truth label, the loss is defined as

$$\mathcal{L} = -\left[y \log p_\theta(\text{yes} \mid P) + (1 - y) \log p_\theta(\text{no} \mid P)\right], \tag{14}$$

where $p_\theta(\cdot \mid P)$ denotes the predicted probability conditioned on prompt $P$ under parameters $\theta$.

## 5 EXPERIMENT

### 5.1 SETTINGS AND BASELINES

We evaluate our method on three widely used benchmarks: FB15k-237 Schlichtkrull et al. (2018), WN18RR Miller (1995), and NELL-995 Carlson et al. (2010), each containing both transductive and inductive subsets. Following prior work Zha et al. (2022); Su et al. (2023; 2024); Li et al. (2025), each query triple is paired with one positive and 49 negative candidate entities for evaluation. We report standard evaluation metrics, including Mean Reciprocal Rank (MRR) and Hit@1 (H@1), which respectively measure the average ranking quality and top-1 prediction accuracy.

Table 1: Transductive and inductive results on WN18RR, FB15k-237, and NELL-995.

| Method | Transductive | | | | | | Inductive | | | | | |
| | WN | | FB15k | | NELL | | WN | | FB15k | | NELL | |
| | H@1 | MRR | H@1 | MRR | H@1 | MRR | H@1 | MRR | H@1 | MRR | H@1 | MRR |
|---|---|---|---|---|---|---|---|---|---|---|---|---|
| RuleN | 0.646 | 0.669 | 0.603 | 0.674 | 0.636 | 0.736 | 0.745 | 0.780 | 0.415 | 0.462 | 0.638 | 0.710 |
| TuckER | 0.600 | 0.646 | 0.615 | 0.682 | 0.729 | 0.800 | - | - | - | - | - | - |
| NCRL | 0.543 | 0.595 | 0.562 | 0.615 | 0.586 | 0.631 | - | - | - | - | - | - |
| GRAIL | 0.644 | 0.676 | 0.494 | 0.597 | 0.615 | 0.727 | 0.769 | 0.799 | 0.390 | 0.469 | 0.554 | 0.675 |
| Adaprop | 0.735 | 0.790 | 0.534 | 0.632 | 0.725 | 0.807 | 0.755 | 0.795 | 0.483 | 0.563 | 0.678 | 0.791 |
| MINERVA | 0.632 | 0.656 | 0.534 | 0.572 | 0.553 | 0.592 | - | - | - | - | - | - |
| BERTRL | 0.655 | 0.683 | 0.620 | 0.695 | 0.686 | 0.781 | 0.755 | 0.792 | 0.541 | 0.605 | 0.715 | 0.808 |
| KRST | 0.835 | 0.899 | 0.639 | 0.720 | 0.694 | 0.800 | 0.809 | 0.890 | 0.600 | 0.716 | 0.649 | 0.769 |
| APST | 0.839 | 0.902 | 0.694 | 0.774 | 0.698 | 0.801 | 0.837 | 0.908 | 0.643 | 0.764 | 0.663 | 0.769 |
| CATS | 0.962 | 0.978 | **0.776** | 0.843 | **0.820** | **0.885** | 0.965 | 0.982 | 0.805 | 0.882 | 0.783 | 0.861 |
| SKILL | **0.962** | **0.979** | 0.774 | **0.845** | 0.789 | 0.865 | **0.971** | **0.984** | **0.859** | **0.911** | **0.839** | **0.903** |

To ensure a fair comparison, we adopt Qwen2-7B-Instruct as the backbone LLM, following the same setting as reported in CATS Li et al. (2025); Zheng et al. (2024). We adopt LoRA Hu et al. (2022) for parameter-efficient fine-tuning, configuring it with a rank of 16 and a scaling factor of 32. The model is optimized using AdamW Loshchilov & Hutter (2019) with a learning rate of 1e-4. We set the per-device batch size to 2 and apply gradient accumulation over 4 steps. The fine-tuning process is conducted for a single epoch. The maximum number of rule premises is set to 3. Each query triple is supplemented with up to 6 closed paths and up to 6 neighboring facts ($K = 6$ in Eq. 10). For instruction construction, 12 negative samples are generated for each positive triple in $T_{\text{train}}$.

To evaluate performance, we benchmark against a comprehensive suite of baselines: embedding-based models (RuleN Meilicke et al. (2018), TuckER Balazevic et al. (2019), NCRL Cheng et al. (2023)), graph neural network-based approaches (GraIL Teru et al. (2020), AdaProp Zhang et al. (2023)), and path or context reasoning models (MINERVA Das et al. (2018), BERTRL Zha et al. (2022), KRST Su et al. (2023), APST Su et al. (2024)), and CATS Li et al. (2025) (current SOTA).

## 5.2 MAIN RESULTS

We evaluate the proposed SKILL framework on three benchmark datasets under both *transductive* and *inductive* settings. The results are shown in Table 1. In the transductive setting, where all entities are observed during training, SKILL achieves competitive performance, obtaining comparable results to state-of-the-art models such as CATS. This demonstrates that even without relying solely on dense KG embeddings, SKILL can effectively leverage structural patterns to make accurate predictions. Notably, SKILL achieves this performance using only half the number of prompts compared to CATS (CATS requires 2 queries for each triple), highlighting its efficiency in extracting and utilizing relevant knowledge.

In the more challenging inductive setting, SKILL consistently outperforms all baseline methods. Notably, it achieves absolute improvements of **5.4%** on FB15k-237 and **5.6%** on NELL-995 in Hit@1, substantially surpassing prior approaches. Unlike prior methods that rely either on static embeddings or unvalidated rules, SKILL explicitly injects semantically validated symbolic rules into LLMs, enabling robust generalization to unseen entities and sparse relational contexts. These gains highlight not only stronger empirical performance but also the methodological novelty of combining symbolic rule mining with LLM-based reasoning, demonstrating the effectiveness of structurally grounded rule injection as a new paradigm for inductive knowledge graph reasoning.

## 5.3 FEW-SHOT RELATION PREDICTIONS

We adopt subsets containing 1000 and 2000 training triplets for all three datasets provided by Zha et al. (2022) to further evaluate SKILL under few-shot settings. Despite the limited scale of training data and the corresponding sparsity of reasoning rules, SKILL exhibits strong generalization performance in the inductive setting.

Table 2: Inductive results in few-shot settings.

| Method | WN-1000 H@1 | WN-1000 MRR | WN-2000 H@1 | WN-2000 MRR | FB15k-1000 H@1 | FB15k-1000 MRR | FB15k-2000 H@1 | FB15k-2000 MRR | NELL-1000 H@1 | NELL-1000 MRR | NELL-2000 H@1 | NELL-2000 MRR |
|---|---|---|---|---|---|---|---|---|---|---|---|---|
| RuleN | 0.649 | 0.681 | 0.737 | 0.773 | 0.207 | 0.236 | 0.344 | 0.383 | 0.282 | 0.334 | 0.418 | 0.495 |
| GRAIL | 0.516 | 0.652 | 0.769 | 0.799 | 0.273 | 0.380 | 0.351 | 0.432 | 0.295 | 0.458 | 0.298 | 0.462 |
| Adaprop | 0.741 | 0.786 | 0.749 | 0.794 | 0.425 | 0.527 | 0.451 | 0.546 | 0.580 | 0.702 | 0.630 | 0.739 |
| BERT-RL | 0.713 | 0.765 | 0.731 | 0.777 | 0.441 | 0.526 | 0.493 | 0.565 | 0.622 | 0.736 | 0.628 | 0.744 |
| KRST | 0.811 | 0.886 | 0.793 | 0.878 | 0.537 | 0.679 | 0.524 | 0.680 | 0.637 | 0.745 | 0.629 | 0.738 |
| APST | 0.822 | 0.894 | 0.798 | 0.879 | 0.561 | 0.697 | 0.627 | 0.747 | 0.654 | 0.765 | 0.637 | 0.747 |
| CATS | 0.864 | 0.922 | 0.923 | 0.953 | 0.776 | 0.862 | 0.802 | 0.877 | **0.713** | **0.808** | **0.746** | **0.829** |
| SKILL | **0.939** | **0.968** | **0.963** | **0.981** | **0.817** | **0.886** | **0.832** | **0.894** | 0.566 | 0.715 | 0.697 | 0.813 |

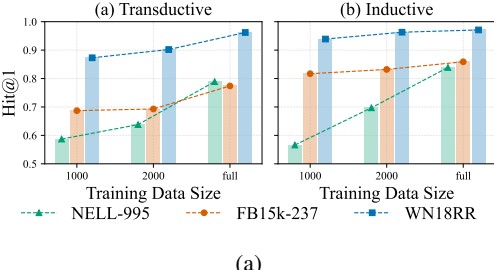
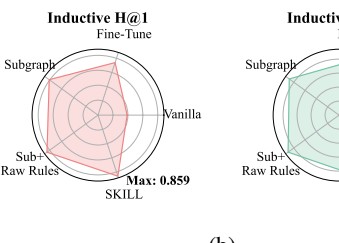

(a)  (b)

Figure 3: (a) Effect of training data scale on reasoning; (b) impacts of SKILL's components.

As shown in Table 2, SKILL achieves up to **7%** absolute improvement over the baselines on two datasets. This substantial gain underscores SKILL's ability to capture and exploit structural patterns in knowledge graphs, even under data-scarce conditions. These results confirm that incorporating symbolic rule guidance into LLMs significantly enhances reasoning by promoting semantically coherent and structurally grounded inferences. Figure 3a further illustrates that increasing the amount of training data leads SKILL to induce more high-quality symbolic rules. The enriched rule set enhances the LLM's understanding of the structural information in the KG, enabling more accurate and reliable reasoning.

### 5.4 ABLATIONS

We evaluate SKILL's components on FB15k-237 under inductive settings, shown in Figure 3b. We compare five variants to assess the impact of SKILL's components. The **vanilla** baseline, which prompts the LLM with raw triples, yields the weakest performance due to the lack of structural context. **Fine-Tune** improves results by training on triples but still lacks relational semantics. **Subgraph** introduces semantically filtered context, yielding clear gains in the inductive setting. Augment-

Table 3: Results on different LLMs.

| Model | Transductive Hit@1 | Transductive MRR | Inductive Hit@1 | Inductive MRR |
|---|---|---|---|---|
| Qwen2 1.5b | 0.714 | 0.802 | 0.778 | 0.854 |
| Qwen2 7b | **0.774** | **0.845** | **0.859** | **0.911** |
| Qwen2.5 7b | 0.752 | 0.826 | 0.824 | 0.890 |
| Llama 3.1 8b | 0.764 | 0.840 | 0.829 | 0.895 |

ing the filtered context with unvalidated symbolic rules (**Sub + Raw Rules**) further improves performance, demonstrating the value of symbolic guidance. The full model, **SKILL**, which combines filtered subgraphs with validated rules, achieves the best results, confirming the complementary strengths of structure-aware context and high-quality rules.

We also evaluate the effects of different LLMs on FB15k-237 dataset, as summarized in Table 3. Even the relatively small **Qwen2 1.5b** achieves competitive results, surpassing traditional rule-based and embedding-based baselines. Scaling to **Qwen2 7b** and **Qwen2.5 7b** further boosts performance, particularly in the inductive scenario where relational generalization is critical. Overall, these results

Table 4: Reaults on the UMLS dataset. Table 5: Rule statistics with different confidence thresholds.

| Method | Hit@1 | MRR |
|--------|-------|-----|
| AMIE | 0.195 | 0.312 |
| Neural-LP | 0.415 | 0.505 |
| RNNLogic | 0.630 | 0.750 |
| NCRL | 0.576 | 0.728 |
| Ruleformer | 0.555 | 0.691 |
| ChatRule | 0.685 | 0.780 |
| **SKILL** | **0.809** | **0.886** |

| Conf | FB15k-237 | | | NELL-995 | | |
|------|-----------|-----|-------|----------|-----|-------|
| | NCRL | Raw | Valid | NCRL | Raw | Valid |
| 0.1 | 32983 | 1613 | 408 | 21469 | 1774 | 691 |
| 0.2 | 29637 | 1237 | 301 | 17841 | 1172 | 466 |
| 0.3 | 27293 | 930 | 205 | 15432 | 818 | 331 |
| 0.4 | 24518 | 709 | 157 | 13916 | 623 | 250 |
| 0.5 | 22371 | 578 | 130 | 12579 | 520 | 214 |

confirm that SKILL can consistently enhance LLMs across different backbones, while even small models already deliver superior performance compared to conventional approaches.

To provide further clarity on model-level differences, Qwen2-7B achieves the strongest overall results among models of similar scale. We attribute this to its post-training stage, which places particular emphasis on logical reasoning, instruction following, and structured knowledge under-standing—capabilities that align closely with symbolic rule–guided inference in SKILL. In contrast, Qwen2.5-7B and Llama 3.1-8B incorporate broader and more diverse training objectives (e.g., cod-ing, multilingual coverage, and general-purpose instruction following), which may introduce small fluctuations in specialized relational reasoning tasks.

To evaluate the generalizability of SKILL beyond open-domain knowledge graphs, we further ex-amine its performance on the biomedical UMLS dataset Kok & Domingos (2007), a widely used domain-specific KG that features specialized relational patterns and medically grounded terminol-ogy. We compare SKILL against representative symbolic and neural rule-learning frameworks, including AMIE Galárraga et al. (2013), Neural-LP Qu et al. (2020), RNNLogic Qu et al. (2020), NCRL Cheng et al. (2023), Ruleformer Xu et al. (2022), and ChatRule Luo et al. (2025).

As shown in Table 4, SKILL achieves the best results on UMLS and surpasses both traditional rule-based systems and recent LLM-enhanced rule learners in terms of Hit@1 and MRR. These findings demonstrate that SKILL transfers effectively to domain-specific relational structures, indicating that the injected structural knowledge remains beneficial even in specialized biomedical settings.

## 5.5 ANALYSIS OF RULES

Table 5 reports the number of induced rules under different confidence thresholds. Although the raw number of candidate rules is large, only a fraction are validated through LLM-based evaluation. By contrast, rules mined by NCRL are overwhelmingly redundant, often yielding tens of thousands of candidates with limited utility, which makes them unsuitable for direct use in downstream reasoning.

More importantly, the validated subset obtained via semantic evaluation preserves rules that are both reliable and useful, thereby reducing redundancy while still covering sufficient reasoning patterns. This filtering effect demonstrates that LLM evaluation not only improves reasoning performance but also prunes spurious or low-quality rules, resulting in a more compact and effective rule base. As illustrated in Figure 3b, despite the substantial reduction in rule count, reasoning accuracy does not decline. Instead, it further improves, underscoring the effectiveness of semantic validation in preserving high-quality rules.

To enable interpretable and semantically grounded reasoning, we extract symbolic rules from the FB15k-237 dataset. These rules are induced from observed multi-hop relational paths and validated via LLM-based semantic prompting to ensure logical plausibility. Table 6 shows representative examples that capture frequent structural patterns in the KG. Such rules act as inductive biases, steering the reasoning process toward explainable predictions.

For instance,

$$\text{speaksLang}(x, y) \ \leftarrow \ \text{actedIn}(x, z) \wedge \text{filmLang}(z, y),$$

encodes an interpretable dependency: if a person acted in a film of a given language, they are likely to speak that language. Such human-readable rules provide both transparency and structural

Table 6: Examples of logical rules from FB15k-237.

| Induced Symbolic Rules |
| --- |
| $\text{hasMajor}(x, y) \leftarrow \text{degreeAt}(x, z) \wedge \text{institutionMajor}(z, y)$ |
| $\text{eventType}(x, y) \leftarrow \text{awardEvent}(x, z) \wedge \text{categoryOf}(z, y)$ |
| $\text{awardEvent}(x, y) \leftarrow \text{categoryOf}(y, z) \wedge \text{eventType}(x, z)$ |
| $\text{speaksLang}(x, y) \leftarrow \text{actedIn}(x, z) \wedge \text{filmLang}(z, y)$ |
| $\text{dubbingLang}(x, y) \leftarrow \text{actedIn}(x, z) \wedge \text{filmLang}(z, y)$ |
| $\text{ethnicGroupLoc}(x, y) \leftarrow \text{hasEthn}(z, x) \wedge \text{nationality}(z, y)$ |
| $\text{nationality}(x, y) \leftarrow \text{hasEthn}(x, z) \wedge \text{ethnicGroupLoc}(z, y)$ |
| $\text{directedFilm}(x, y) \leftarrow \text{founded}(z, x) \wedge \text{prodCompany}(y, z)$ |

grounding, guiding LLM reasoning toward reliable predictions. Appendix G presents a detailed case study that further highlights the interpretability and reliability of SKILL.

## 6 CONCLUSION

We presented **SKILL**, a framework for inductive knowledge graph reasoning that integrates symbolic knowledge into large language models. By leveraging one-shot prompting to extract and validate path-based rules, SKILL filters out noisy patterns and injects high-quality relational knowledge to guide the reasoning process. This design introduces a novel paradigm that combines the interpretability of symbolic rule mining with the generalization ability of LLMs. Experiments on three standard benchmarks demonstrate consistent gains over state-of-the-art methods, achieving up to 5 absolute improvements in Hit@1. In future work, we plan to extend SKILL to open-world scenarios and explore its applicability to broader tasks, such as knowledge-based question answering and graph-based recommendations.

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

## A  ALGORITHMS OF SKILL

Algorithm 1 outlines a two-stage framework for mining logical rules from knowledge graphs by integrating statistical patterns with semantic validation.

In the first stage, the algorithm traverses paths between entity pairs using a breadth-first search up to a specified length $k$, generating candidate rules from the observed paths. Each rule $\rho$ is evaluated based on its empirical confidence in the KG, and retained if it exceeds a threshold $\tau$, forming the raw rule set $R_{\text{raw}}$. This step ensures that the selected rules exhibit statistically meaningful patterns.

In the second stage, each rule in $R_{\text{raw}}$ is assessed using a large language model to verify its semantic plausibility. Rules that pass this LLM-based validation are retained in the final set $R_{\text{valid}}$. This filtering mechanism helps eliminate spurious or logically inconsistent rules that may arise from purely statistical associations.

To distinguish induced rules from the original relation set $\mathcal{R}$ of the knowledge graph $\mathcal{G} = (\mathcal{E}, \mathcal{R}, \mathcal{T})$, the output rule set is denoted as $R$. This hybrid approach ensures that the resulting symbolic rules are both statistically robust and semantically coherent, providing reliable inductive bias for downstream reasoning tasks.

---

**Algorithm 1** Rule-Miner

---

**Require:** Knowledge graph $\mathcal{G} = (\mathcal{E}, \mathcal{R}, \mathcal{T})$, path length $k$
**Ensure:** Set of valid rules $R_{\text{valid}}$
 1: $R_{\text{raw}} \leftarrow \emptyset$
 2: **for all** triples $(h, r, t) \in \mathcal{T}$ **do**
 3:    $\Pi(h, t) \leftarrow \text{BFS}(h, t, k, \mathcal{G})$
 4:    **for all** path $\pi \in \Pi(h, t)$ **do**
 5:        $\rho \leftarrow \text{ExtractRule}(\pi, r)$
 6:        **if** $\text{ComputeConfidence}(\rho, \mathcal{G}) > \tau$ **then**
 7:            $R_{\text{raw}} \leftarrow R_{\text{raw}} \cup \{\rho\}$
 8: $R_{\text{valid}} \leftarrow \emptyset$
 9: **for all** $\rho \in R_{\text{raw}}$ **do**
10:    **if** $\text{IsPlausibleLLM}(\rho, \mathcal{G})$ **then**
11:        $R_{\text{valid}} \leftarrow R_{\text{valid}} \cup \{\rho\}$
12: **return** $R_{\text{valid}}$

---

**Logic-Enhanced Reasoning** aims to improve the inductive reasoning capabilities of LLMs by incorporating validated symbolic rules into the learning process. Given a query triple $(h, r, t)$, SKILL

first constructs a reasoning subgraph that integrates both the first-order neighborhoods and multi-hop relational paths connecting the head and tail entities.

To reduce noise and emphasize structurally meaningful evidence, a rule-matching mechanism is employed to align relational paths in the subgraph with high-confidence symbolic rules. This process yields a pruned subgraph and a set of relevant rules, both of which are transformed into natural language prompts that encode rich structural and semantic context. These prompts are then used to fine-tune the LLM, enabling it to assess the plausibility of candidate triples. By explicitly injecting relational structure and symbolic knowledge, SKILL facilitates generalization to unseen entities and sparse subgraphs while maintaining interpretability through rule-grounded reasoning.

---

**Algorithm 2** Reasoning Subgraph Construction

---

**Require:** Knowledge graph $\mathcal{G} = (\mathcal{E}, \mathcal{R}, \mathcal{T})$, query triple $(h, r, t)$, max path length $k$
**Ensure:** Reasoning subgraph $\mathcal{G}_{(h,r,t)}$
1: $\mathcal{N}_1(h) \leftarrow \{(h, r', e') \in \mathcal{T}\} \cup \{(e', r', h) \in \mathcal{T}\}$
2: $\mathcal{N}_1(t) \leftarrow \{(t, r', e') \in \mathcal{T}\} \cup \{(e', r', t) \in \mathcal{T}\}$
3: $\mathcal{P}_{h \to t} \leftarrow \text{BFS}(h, t, k, \mathcal{G})$
4: **return** $\mathcal{G}_{(h,r,t)} = \mathcal{N}_1(h) \cup \mathcal{N}_1(t) \cup \bigcup_{\pi \in \mathcal{P}_{h \to t}} \text{Triples}(\pi)$

---

As detailed in Algorithm 2, the reasoning subgraph $\mathcal{G}_{(h,r,t)}$ is composed of two components: (i) the first-order neighborhoods of the head and tail entities, capturing local relational context, and (ii) closed multi-hop relational paths of bounded length (up to $k$) between the head and tail entities, extracted using a breadth-first search. Together, these elements provide a rich and contextually grounded subgraph that serves as the foundation for logic-informed reasoning.

To reduce noise and highlight informative patterns, we prune the reasoning subgraph using a symbolic strategy (Algorithm 3). For each closed path $\pi$ in $\mathcal{G}_{(h,r,t)}$ and rule $\rho \in R$, we compute a soft match score based on how well $\pi$ aligns with the body of $\rho$. This score is weighted by the rule's confidence, yielding a relevance score $s(\pi, \rho)$.

---

**Algorithm 3** Reasoning Subgraph Pruning

---

**Require:** Reasoning subgraph $\mathcal{G}_{(h,r,t)}$, validated rules $R$ with confidence $\text{conf}(\rho)$, top-$K$ size $K$
**Ensure:** Pruned subgraph $\tilde{\mathcal{G}}_{(h,r,t)}$, relevant rules $R_{(h,r,t)}$
1: Initialize list $S \leftarrow []$
2: **for all** closed paths $\pi$ in $\mathcal{G}_{(h,r,t)}$ **do**
3:     **for all** rules $\rho \in R$ **do**
4:         $m(\pi, \rho) \leftarrow$ fraction of rule body matched by $\pi$
5:         $s(\pi, \rho) \leftarrow m(\pi, \rho) \times \text{conf}(\rho)$
6:         **if** $s(\pi, \rho) > 0$ **then**
7:             Append $(\pi, \rho, s(\pi, \rho))$ to $S$
8: $\mathcal{S}_{(h,r,t)} \leftarrow \text{TopK}(S, K)$
9: $\tilde{\mathcal{G}}_{(h,r,t)} \leftarrow \bigcup_{(\pi, \rho, s) \in \mathcal{S}_{(h,r,t)}} \text{Triples}(\pi)$
10: $R_{(h,r,t)} \leftarrow \{\rho \mid \exists \pi : (\pi, \rho, s) \in \mathcal{S}_{(h,r,t)}\}$
11: **return** $\tilde{\mathcal{G}}_{(h,r,t)}, R_{(h,r,t)}$

---

We rank all path-rule pairs by relevance and select the top-$K$ candidates. The resulting pruned subgraph $\tilde{\mathcal{G}}_{(h,r,t)}$ and associated rule set $R_{(h,r,t)}$ retain the most semantically aligned evidence, improving both reasoning focus and interpretability. After pruning, the refined reasoning subgraph $\tilde{\mathcal{G}}(h, r, t)$ and the corresponding rule set $R(h, r, t)$ are converted into a natural language prompt that encodes both structural context and symbolic guidance for the query triple.

## B STATISTICS OF DATASETS

We conduct experiments on three widely used benchmark datasets: FB15k-237, WN18RR, and NELL-995. **FB15k-237** is a subset of Freebase with redundant inverse relations removed, com-

monly used for link prediction. **WN18RR** is derived from WordNet, a lexical knowledge base, and retains only non-trivial relations to avoid test leakage. **NELL-995** originates from the Never-Ending Language Learning system, containing automatically extracted facts with a larger and noisier structure compared to the other two datasets. These datasets collectively cover diverse domains—encyclopedic knowledge, lexical semantics, and open-domain extractions—providing a comprehensive testbed for evaluating inductive reasoning methods. Statistics of FB15k-237, WN18RR, and NELL-995 datasets are summarized in Table 7. The implementation code and datasets are provided in the supplementary material.

Table 7: Statistics of datasets and their splits. $|\mathcal{R}_G|$: #relations, $|\mathcal{E}_G|$: #entities, $|\mathcal{T}_G|$: #triplets.

| Split | FB15k-237 | | | WN18RR | | | NELL-995 | | |
|---|---|---|---|---|---|---|---|---|---|
| | $|\mathcal{R}_G|$ | $|\mathcal{E}_G|$ | $|\mathcal{T}_G|$ | $|\mathcal{R}_G|$ | $|\mathcal{E}_G|$ | $|\mathcal{T}_G|$ | $|\mathcal{R}_G|$ | $|\mathcal{E}_G|$ | $|\mathcal{T}_G|$ |
| train | 180 | 1594 | 5223 | 9 | 2746 | 6670 | 88 | 2564 | 10063 |
| train-2000 | 180 | 1280 | 2008 | 9 | 1970 | 2002 | 88 | 1346 | 2011 |
| train-1000 | 180 | 923 | 1027 | 9 | 1362 | 1001 | 88 | 893 | 1020 |
| test-transductive | 102 | 550 | 492 | 7 | 962 | 638 | 60 | 1936 | 968 |
| test-inductive | 142 | 1093 | 2404 | 8 | 922 | 1991 | 79 | 2086 | 6621 |

## C PROMPT TEMPLATE DESIGN

---
**Prompt (Evaluating the plausibility of rules)**

You are an expert in knowledge reasoning and rule-based inference. Your task is to evaluate the following reasoning rule and its instance.
Your evaluation should consider two aspects:
**1. Reasonableness:**

- Does the rule logically follow from known facts or principles?
- Are the premises valid and do they logically support the conclusion?
- Is there sufficient evidence to justify this inference?
- Is the rule premise the same? If so, the rule is not reasonable.

**2. Usefulness:**

- Can this rule be applied in practical real-world scenarios?
- Can it contribute to meaningful inference or prediction?
- Does it help to reduce uncertainty, assist decision making, or generate new knowledge?

**Decision Criteria:** If the rule is both reasonable and useful, answer "Yes". If the rule fails to meet either Reasonableness or Usefulness, answer "No".
**Example Evaluation:** *Rule Head:* person gender *Rule Premise:* person spouse s marriage type of union. person spouse s marriage type of union. person gender
*Explanation:* This rule tries to infer a person's gender based on having a spouse and knowing the marriage type of union. However, knowing someone is married and the marriage type does not allow inference of gender. Therefore, this rule is not reasonable or useful. *Answer:* No
**Evaluation Task:** Rule Head: {rule head} Rule Premise: {rule body} Rule Instance: Result triple: {result triple} Premise triple: {premises triples}
Please answer only with "Yes" or "No". Do not provide any additional explanation or context.
---

The prompt template used to evaluate the plausibility of rules is shown in Prompt (Evaluating the plausibility of rules). It guides the language model to assess each rule-instance pair based on two criteria: *Reasonableness* and *Usefulness*. The model is instructed to examine whether the rule logically follows from its premises and whether it can contribute to practical inference or decision-

making. A binary decision—"Yes" or "No"—is then returned, indicating whether the rule is both logically sound and pragmatically valuable.

The prompt template used to evaluate whether a specific relation can be reliably inferred from the knowledge graph is shown in Prompt (Inference Verification). It instructs the LLM to consider a combination of local neighbor triples, inductively derived reasoning rules, and closed relational paths connecting the head and tail entities. Based on structured context, the model determines whether the target relation is inferable.

---

**Prompt (Inference Verification)**

You are an expert in knowledge reasoning. Your task is to determine whether the relation in the input can be reliably inferred between the head and tail entities, based on a set of reasoning paths from the knowledge graph.
The head entity is {head_entity}, the tail entity is {tail_entity}.
**Neighbor triples from the knowledge graph:**
{neighbor_triples}
**Reasoning rules inductively derived from the graph:**
{reasoning_rules}
**Closed paths collected of the knowledge graph:**
{reasoning_paths}
**The relation to be inferred:**
{test_triple}
Please return "Y" if the triplet can be inferred from the knowledge graph based on the reasoning paths and rules provided, otherwise return "N". Do not say anything else except your determination.

---

## D   DETAILED TRAINING IMPLEMENTATION

We provide a comprehensive overview of the training procedure in SKILL, covering offline rule mining, dynamic prompt construction, and optimization settings.

**Offline Rule Mining and Verification** The workflow begins with an offline preprocessing stage that constructs a repository of semantically validated rules. Candidate rules are first generated using a breadth-first search (BFS) over the training graph to identify closed-path symbolic patterns. Each candidate is then evaluated via one-shot prompting with a Large Language Model (LLM). Only rules deemed "reasonable" and "useful" are retained in the validated rule set $\mathcal{R}_{valid}$, effectively removing spurious or noisy patterns before training.

**Dynamic Prompt Construction** For each training triple $(h, r, t)$, we build a structured prompt by integrating subgraph evidence with the most relevant validated rules. We first retrieve the local reasoning subgraph $\mathcal{G}_{(h,r,t)}$, which includes the first-order neighborhood and closed paths connecting $h$ and $t$. Each rule in $\mathcal{R}_{valid}$ is then scored according to how well its body matches the extracted paths (Eq. 8). The Top-$K$ (with $K = 6$) highest-scoring rule–path pairs are selected to form the final instruction prompt. To enable discriminative learning, we also construct 12 negative instances for every positive triple by corrupting its head or tail entity.

**Hyperparameters and Optimization** We fine-tune the `Qwen2-7B-Instruct` backbone using LoRA for parameter-efficient adaptation, with rank $r = 16$ and scaling factor $\alpha = 32$. Optimization is performed using AdamW with a learning rate of $1 \times 10^{-4}$. The effective batch size is controlled by combining a per-device batch size of 2 with gradient accumulation over 4 steps. To avoid overfitting, training is carried out for a single epoch.

**Inference Workflow** During inference, each test triple undergoes the same prompt construction procedure as during training. The model receives the retrieved reasoning subgraph and the Top-6 matched rules, and predicts the plausibility of the query by outputting a "Yes" or "No" response.

Regarding runtime, we report the measurements of our own implementation. With the closed-path constraint, rule mining completes within 20 minutes on all datasets, and LLM-based semantic validation proceeds at roughly 4 rules per second. These steps are fully offline and do not affect inference

latency. At inference time, SKILL processes about 5 queries per second, which is approximately 50% of the throughput reported by CATS.

# E    HUMAN EVALUATION OF LLM-BASED RULE VALIDATION

To assess the reliability of the LLM-based rule validation stage, we conduct a human examination of a representative set of **100 validated rules** sampled from the rule base constructed on FB15k-237. Each rule is reviewed for semantic coherence and reasonableness within the Freebase schema.

Our analysis shows that approximately **87%** of the validated rules are judged semantically coherent or at least plausible, whereas around **13%** are found to be implausible. This suggests that the LLM-based semantic validation is reasonably reliable in practice.

Most rules capture sensible multi-hop dependencies such as *film→distributor→genre* or *degree→institution→major*. The small portion of implausible rules is further mitigated by our match–confidence scoring and Top-$K$ pruning during inference. Overall, this human study shows that LLM-based rule validation produces a compact and mostly accurate rule set that effectively supports structural reasoning in our framework.

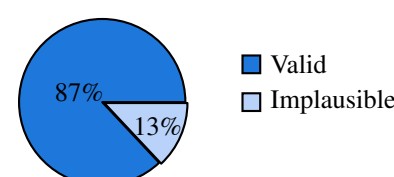

Figure 4: Human judgment distribution over 100 validated rules.

# F    RESULTS OF ABLATION STUDIES

We evaluate SKILL's components on FB15k-237 under both transductive and inductive settings, as shown in Table 8. The **Vanilla** baseline directly prompts the LLM with textualized triples, yielding the weakest performance due to the lack of adaptation and structural cues. **Fine-Tune** improves results by training on raw triples, but still lacks explicit relational understanding. **Subgraph** only introduces semantically filtered subgraphs, leading to notable gains, especially in the inductive setting, by providing structured contextual information. Adding unfiltered symbolic rules (**Sub + Raw Rules**) further boosts performance, indicating that rule-level guidance is beneficial even without validation. **SKILL** achieves the best results by combining filtered subgraphs with validated rules, showing the importance of both semantic filtering and symbolic guidance. Overall, these results highlight the complementary roles of structure-aware subgraph pruning and high-quality symbolic rules in enhancing LLM-based knowledge graph reasoning. Although SKILL yields only a moderate improvement over **Sub + Raw Rules**, it relies on a substantially pruned set of rules that are fewer in number but higher in quality, ensuring that the performance gains are both more reliable and more consistent.

Table 8: Ablation results on FB15k-237

| Conf. | Transductive | | Inductive | |
|---|---|---|---|---|
| | H@1 | MRR | H@1 | MRR |
| Vanilla | 0.461 | 0.564 | 0.390 | 0.508 |
| Fine-Tune | 0.723 | 0.813 | 0.739 | 0.832 |
| Subgraph | 0.751 | 0.839 | 0.807 | 0.879 |
| Sub + Raw Rules | 0.763 | 0.837 | 0.844 | 0.904 |
| SKILL | **0.774** | **0.845** | **0.859** | **0.911** |

# G    CASE STUDY

To further demonstrate the interpretability and reliability of **SKILL**, we present a case study in the education domain (see Fig. 5). The target query is to infer whether *Barack Obama*'s major is *Law*. Unlike conventional embedding-based methods that rely on latent similarity and may traverse semantically implausible paths, SKILL explicitly grounds inference in symbolic rules validated by

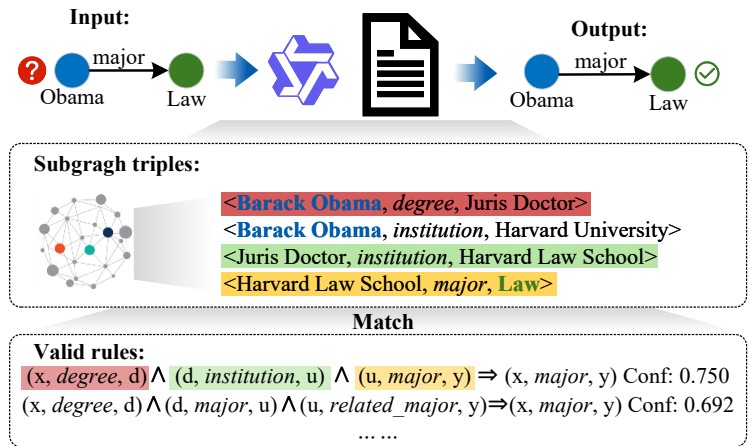

Figure 5: Case study: SKILL infers `<Obama, major, Law>` via explicit symbolic reasoning, providing both correct prediction and transparent interpretability.

LLMs. From the local subgraph, SKILL collects supporting triples `<Obama, degree, Juris Doctor>`, `<Juris Doctor, institution, Harvard Law School>`, and `<Harvard Law School, major, Law>`. Matching these facts with a validated rule,

$$(x, \text{degree}, d) \wedge (d, \text{institution}, u) \wedge (u, \text{major}, y) \Rightarrow (x, \text{major}, y),$$

SKILL infers the missing relation `<Obama, major, Law>`. This explicit, rule-aligned chain provides a transparent explanation for the prediction and illustrates how symbolic guidance improves the semantic consistency of LLM-based reasoning.

## H  THE USE OF LARGE LANGUAGE MODELS (LLMS)

During the preparation of this paper, LLMs are used exclusively for language polishing and stylistic refinement. Specifically, LLMs improve the clarity, fluency, and readability of the manuscript without altering its substantive content, methodology, or results. All conceptual ideas, experimental designs, data analyses, and conclusions are conceived and executed independently by the authors. The use of LLMs is therefore limited to surface-level improvements, such as correcting grammar, adjusting phrasing for conciseness, and ensuring consistency in academic tone, in order to meet the standards of formal scientific writing.

