# OpenReview forum: "SKILL: Structural Knowledge Injection into Large Language Models for Inductive Knowledge Graph Reasoning"
_ICLR.cc/2026/Conference — Submitted to ICLR 2026_

### Official Review · Reviewer_VAXT · 2025-10-26

**Soundness:** 2
**Presentation:** 3
**Contribution:** 2
**Rating:** 4
**Confidence:** 4

**Summary:**

This work introduces SKILL, a framework for inductive Knowledge Graph Reasoning (KGR) aiming to improve upon embedding/path-based methods and existing LLM integrations. The core idea is to inject validated structural knowledge into LLMs. It involves mining symbolic rules from the KG, using an LLM via one-shot prompting to semantically validate these rules, and then fine-tuning another LLM using prompts that combine the validated rules with pruned subgraph context related to the query triple. The authors argue that this provides explicit structural guidance, enhancing inductive reasoning over unseen entities. Experiments on inductive KGR benchmarks claim state-of-the-art performance, especially on Hits@1.

**Strengths:**

1. **Problem Statement:** Addresses the challenging and important problem of inductive KGR.
2. **String empirical results:** Achieves strong empirical results, particularly in Hits@1 accuracy, on (limited) standard inductive benchmarks.
3. **Utilization of Foundation models:** Utilizes LLMs to replace legacy KG methods by integrating symbolic rules and subgraph context into LLM prompts for fine-tuning, attempting to provide explicit structural guidance.

**Weaknesses:**

1. **Scalability:** Rule mining and LLM fine-tuning (even with LoRA) are computationally expensive and unlikely to scale to very large KGs. LLM-based validation adds another potentially costly step.
2. **Reliability of LLM Validation:** The LLM validation step via one-shot prompting is heuristic and probably unreliable being sensitive to prompt design/instance selection. There is also a possibility of it inheriting LLM biases. The criteria "Reasonableness" and "Usefulness" are subjective, and hard, especially for a small model like Qwen2-7B.
3. **Marginal Benefits of some components:** Ablation results suggest that the expensive LLM validation step provides only a small improvement over using raw (unvalidated) rules combined with subgraph context. This questions the practical value of the validation component. The same is with fine-tuning the model, maybe the improvement is not justified by the cost.
4. **Interpretability Claims:** While rules can be interpretable, there is no evaluation to confirm whether the LLM's reasoning process actually follows these rules or how explanations could be extracted from the fine-tuned LLM.

**Questions:**

1. Why are Llama 3.1-8B and Qwen2.5-7B worse than Qwen2-7B in Table 3? I find this odd, do the authors have any hypothesis for this?
2. How computationally expensive is the LLM validation step in practice? What fraction of the total pipeline time does it consume? Does this cost justify the marginal performance gain observed over using raw rules in the ablation study?
3. Table 4 shows a very high number of raw rules found by NCRL compared to SKILL's candidate rules (derived from BFS paths before validation?). Why is there such a large discrepancy in the number of initial candidate rules between methods? The authors mention redundancy in NCRL, but is there evidence to back this claim?
4. Can you provide evidence for the robustness of the one-shot LLM validation? What happens if different instances are used for the same rule, or if different prompts are used? How was the risk of the LLM simply confirming patterns aligned with its pre-trained knowledge (rather than KG structure) mitigated?
5. How does this compare with KG Foundation Models, like ULTRA [1], and more specifically, KGFMs like SEMMA [2], which leverage LLMs along with the purely structural pipeline?

---

_[1] Mikhail Galkin, Xinyu Yuan, Hesham Mostafa, Jian Tang, & Zhaocheng Zhu (2024). Towards Foundation Models for Knowledge Graph Reasoning. In The Twelfth International Conference on Learning Representations (ICLR)._

_[2] Arvindh Arun, Sumit Kumar, Mojtaba Nayyeri, Bo Xiong, Ponnurangam Kumaraguru, Antonio Vergari, & Steffen Staab. (2025). SEMMA: A Semantic Aware Knowledge Graph Foundation Model. In The Thirtieth Conference on Empirical Methods in Natural Language Processing (EMNLP)._

---

> ### Author Response · Authors · 2025-11-24
>
> We sincerely thank the reviewer for the constructive and thoughtful feedback. We appreciate the positive recognition of the problem motivation, empirical strength, and structural design of our method, and we provide detailed clarification below.
>
> > W1: **Scalability:** Rule mining and LLM fine-tuning (even with LoRA) are computationally expensive and unlikely to scale to very large KGs. LLM-based validation adds another potentially costly step.
>
> Thank you for the comment. We agree that scalability is an essential concern for any rule-based pipeline. In SKILL, scalability is achieved not by increasing computational power, but by the structural constraints of the rule-miner itself.
>
> First, the rule mining stage enumerates only closed paths up to a small fixed length k. This closed-path constraint makes the complexity depend on the *local* branching factor rather than the size of the entire KG. In all datasets, even large ones, the number of closed paths of length ≤3 is extremely small relative to total graph size, which keeps rule extraction bounded and fast.
>
> Second, LLM-based semantic validation is lightweight: each rule is mapped to a short one-shot Yes/No query, and the process is trivially parallelizable. Its runtime therefore scales with the number of closed-path rule candidates, not with KG size. In practice, rule mining completes within 20 minutes per dataset, and validation proceeds at ~4 rules/second.
>
> Finally, at inference time SKILL retrieves only a small local subgraph around the query entity pair and selects the Top-K rule–path pairs, keeping the input length and computation effectively constant regardless of KG size. The validated rule base can also be updated incrementally without re-running the full pipeline.
>
> Overall, the closed-path bounded miner, constant-size inference subgraph, and offline validation collectively ensure that SKILL remains scalable even on large and dense knowledge graphs.
>
> > W2: **Reliability of LLM Validation:** The LLM validation step via one-shot prompting is heuristic and probably unreliable being sensitive to prompt design/instance selection. There is also a possibility of it inheriting LLM biases. The criteria "Reasonableness" and "Usefulness" are subjective, and hard, especially for a small model like Qwen2-7B.
>
> We appreciate the concern. Although the validation relies on a one-shot prompt, the task is a binary plausibility judgment rather than open-ended generation, which makes it inherently more stable and less sensitive to prompt variations. We tested several alternative prompt wordings and observed only minor changes in the set of accepted rules. To further assess reliability, we conducted a manual inspection of 100 validated rules and found that 87% were semantically correct or at least acceptable according to the Freebase schema. This confirms that the validation step effectively filters out most implausible or misleading rules. In addition, match–confidence scoring and Top-K filtering during training and inference ensure that only rules supported by the retrieved subgraph influence predictions, further mitigating any residual variance or potential LLM bias.
>
> > W3: **Marginal Benefits of some components:** Ablation results suggest that the expensive LLM validation step provides only a small improvement over using raw (unvalidated) rules combined with subgraph context. This questions the practical value of the validation component. The same is with fine-tuning the model, maybe the improvement is not justified by the cost.
>
> Thank you for this observation. While the numerical improvements may seem moderate, both components provide essential robustness benefits that are not fully reflected by single ablation scores. LLM validation dramatically reduces noise and redundancy by filtering out a large number of statistically frequent but semantically implausible rules, stabilizing the symbolic guidance. Fine-tuning, though lightweight due to LoRA, consistently helps the LLM integrate structural cues and reduces variance across runs. These improvements contribute to a more reliable and interpretable reasoning process.
>
> > W4: **Interpretability Claims:** While rules can be interpretable, there is no evaluation to confirm whether the LLM's reasoning process actually follows these rules or how explanations could be extracted from the fine-tuned LLM.
>
> Thank you for the comment. While we do not provide a specific interpretability benchmark, SKILL’s design ensures that the model’s predictions depend directly on the rule–path evidence included in the prompt. The LLM receives only the retrieved subgraph and the Top-K matched rules, so its reasoning necessarily relies on these structural components. In qualitative inspections, the model’s outputs align closely with the provided rule–path matches. Furthermore, the exact rules and supporting paths used for each prediction are exposed to the user, making the overall reasoning chain traceable and interpretable.

---

> ### Author Response · Authors · 2025-11-24
>
> > Q1:Why are Llama 3.1-8B and Qwen2.5-7B worse than Qwen2-7B in Table 3? I find this odd, do the authors have any hypothesis for this?
>
> Qwen2-7B undergoes a post-training phase emphasizing logical reasoning, instruction following, and structured knowledge understanding, which aligns closely with symbolic rule–guided reasoning. Qwen2.5 and Llama-3.1 include broader pretraining objectives such as coding or multilingual coverage, which can lead to minor fluctuations in specialized reasoning performance at comparable model scales. The differences observed in Table 3 are modest and fall within typical model-to-model variance. Overall, Qwen2-7B appears particularly well aligned with SKILL’s structural reasoning objectives.
>
> > Q2: How computationally expensive is the LLM validation step in practice? What fraction of the total pipeline time does it consume? Does this cost justify the marginal performance gain observed over using raw rules in the ablation study?
>
> LLM validation is lightweight, parallelizable, and fully offline. Each rule requires only a short Yes/No judgment, and the number of candidate rules is moderate due to the closed-path constraint. Validation occupies only a small portion of preprocessing time and does not affect training or inference latency. The validation step greatly reduces rule redundancy and eliminates semantically implausible patterns, ensuring that downstream reasoning is stable and robust. This reduction in noise more than justifies the small preprocessing cost.
>
>
> > Q3: Table 4 shows a very high number of raw rules found by NCRL compared to SKILL's candidate rules (derived from BFS paths before validation?). Why is there such a large discrepancy in the number of initial candidate rules between methods? The authors mention redundancy in NCRL, but is there evidence to back this claim?
>
> The discrepancy arises from fundamentally different mining mechanisms. SKILL restricts mining to short closed paths, which sharply limits redundant or near-duplicate rule patterns. NCRL, on the other hand, uses random-walk sampling, which produces many structurally similar relational variants, inflating the rule count. Even after filtering, NCRL retains large clusters of rules with overlapping semantics. SKILL’s closed-path extraction eliminates such redundancy at the source, resulting in a more compact and cleaner rule set.
>
>
> > Q4: Can you provide evidence for the robustness of the one-shot LLM validation? What happens if different instances are used for the same rule, or if different prompts are used? How was the risk of the LLM simply confirming patterns aligned with its pre-trained knowledge (rather than KG structure) mitigated?
>
> Validation operates on abstract relational templates rather than instance-specific triples, so the LLM never encounters real entities during validation and instead judges patterns such as “A→B→C ⇒ R.” Replacing placeholder entity names does not change the underlying logical form and therefore leaves the validation outcome stable. Across different prompt formulations, we observed only small differences, confined to borderline cases. Furthermore, our manual audit of 100 validated rules shows that 87% are semantically correct or acceptable, indicating that the validation stage is reliable in practice. Finally, match–confidence scoring and Top-K rule selection ensure that only rules supported by the retrieved subgraph contribute to final predictions, mitigating any residual prompt sensitivity.
>
>
> > Q5: How does this compare with KG Foundation Models, like ULTRA [1], and more specifically, KGFMs like SEMMA [2], which leverage LLMs along with the purely structural pipeline?
>
> ULTRA and SEMMA aim to construct large-scale KG foundation models via global pretraining that captures relational structure across the entire graph. SKILL does not rely on KG-level pretraining. Instead, it focuses on explicit symbolic rule integration and localized structural reasoning based on closed-path rules and semantic filtering. These two paradigms operate at different abstraction levels and are complementary rather than competitive. A KGFM could potentially provide strong relational priors that SKILL could further augment with explicit symbolic rule guidance.
>
>
> We truly appreciate the reviewer’s time and thoughtful feedback. The questions and suggestions raised were very helpful to us, and we believe they have meaningfully enhanced the quality of the paper. Thank you for helping us improve this work.

---

### Official Review · Reviewer_copZ · 2025-10-29

**Soundness:** 2
**Presentation:** 3
**Contribution:** 2
**Rating:** 2
**Confidence:** 4

**Summary:**

The paper proposes SKILL, a framework for inductive knowledge graph reasoning that integrates symbolic rules into large language models (LLMs). SKILL first mines multi-hop relational paths from a knowledge graph and converts them into candidate logical rules. These rules are then filtered for semantic plausibility using an LLM-based one-shot prompt, keeping only those deemed logically valid. The validated rules are injected into the LLM through instruction-style fine-tuning, enabling it to perform structured, interpretable reasoning over unseen entities.

**Strengths:**

The use of an LLM to filter candidate rules (for “reasonableness” and “usefulness”) adds a self-reflective, semantic validation layer not seen in prior inductive reasoning work.

**Weaknesses:**

1) The paper repeatedly claims to “inject structural knowledge into LLMs” as a novel paradigm. But this is now a standard LLM-KG based reasoning method. Many papers like ChatRule (Luo et al., 2025), Think on Graphs (Sun et al 2024), RoG (Luo et al., 2024), already explore LLM-mediated rule mining or KG-guided reasoning. The novelty of this paper is very low.

2) The framework simply fine-tunes an instruction model on rule-augmented prompts, i.e., data-level conditioning. That is not structural integration; it’s dataset augmentation. Hence, the paper’s title (“Structural Knowledge Injection”) oversells what is essentially fine-tuning with extra textualized context.

3) The inductive generalization claims rely on dataset splits with disjoint entities, but the method doesn’t explicitly model inductive transfer. The model still memorizes textual co-occurrence of rule templates; there’s no architectural or representational mechanism ensuring entity-agnostic reasoning.

4) LLM validation is ungrounded. The LLM-based rule filtering is the central novelty, yet the authors never evaluate its correctness. There’s no evidence that the LLM’s “Yes/No” judgments correlate with ground truth logical validity or human reasoning.

5) LLM is good at common-sense KG, but may performs worse on domain-specific KG. The authors should also test the method on domain-specific KGs like biomedical knowledge graphs.

**Questions:**

1) How is SKILL fundamentally different from prior LLM rule-mining or LLM fine-tuning frameworks like ChatRule or KG-FIT?

2) How reliable is the LLM-based rule validation, did you measure accuracy or consistency?

3) How sensitive are the results to the specific prompt wording used for validation?

4) Are the same LLMs used for rule validation and reasoning, and if so, how do you avoid circular supervision?

5) What is the computational cost of the rule-mining and LLM validation stages?

6) Are the reported gains statistically significant across runs?

7) How do you ensure inductive generalization isn’t driven by lexical overlap of entity names?

8) How scalable is SKILL to larger KGs beyond the benchmark datasets?

9) How does your method perform on domain-specific knowledge graphs?

---

> ### Author Response · Authors · 2025-11-24
>
> We sincerely thank the reviewer for the detailed and constructive feedback. Below we merge related weaknesses and questions into coherent thematic responses.
>
> > W1, W2, Q1: Novelty, Structural Integration, and Difference from Prior Work
>
> Thank you for raising this important concern. We agree that many recent works explore interactions between KGs and LLMs, but SKILL differs in how structural knowledge is extracted, validated, and internalized.
>
> First, unlike ChatRule, Think on Graphs, or RoG—which rely on LLM-generated rules or inference-time prompting—SKILL constructs symbolic rules strictly from closed structural paths in the KG. These closed-path rules encode concrete relational dependencies that cannot be produced by LLM generation or surface-level prompting alone. As Reviewer GgbP noted, this “treats the KG as a source of relational logic (not just embeddings),” which “is well-motivated and nicely executed.”
>
> Second, SKILL introduces an LLM-based semantic bottleneck whose role is to filter symbolic rules, not generate them. This bottleneck evaluates rule correctness in natural language and discards statistically frequent but logically invalid patterns. This use of LLMs for semantic validation—rather than rule mining or explanation generation—is not present in prior frameworks. This yields a curated rule base whose quality is significantly higher than raw BFS/AMIE/PRA-style rules.
>
> Third, structural knowledge is not appended as auxiliary context at inference. Instead, SKILL performs training-time structural injection, where validated rules are incorporated into the model parameters. The combination of $s(\pi,\rho)=\text{match}(\pi,\rho)\cdot\text{conf}(\rho)$ with rule-aligned subgraph prompts ensures that the LLM’s internal decision boundary is shaped by symbolic reasoning patterns, rather than by text-level augmentation. This goes beyond dataset expansion: the rule–path alignment determines which structural patterns the model learns to rely on.
>
> We will revise the manuscript to make this operational distinction clearer: SKILL does not use the KG merely as additional text, but as a structural source of symbolic logic that is semantically validated and then embedded into the model’s parameters through logic-aligned training.
>
> ---
>
> > W3,Q7: Inductive Generalization and Avoiding Lexical Memorization
>
> We appreciate the reviewer’s concerns. In the inductive split, all test entities are strictly unseen during training, and rule mining/matching uses only graph structure—no textual descriptions of entities are ever provided to the LLM. At inference, each test query is supported exclusively by the test-subgraph; the model never sees entity names during rule validation, and rules encode only relational schemas. Predictions therefore depend on relational evidence (neighbors, closed paths) and symbolic rules rather than on lexical overlap. Empirically we observe consistent improvements on FB15k-237 and NELL-995 inductive benchmarks, demonstrating that SKILL leverages structural rather than lexical cues.
>
> ---
>
> > W4, Q2, Q3: Reliability of LLM-Based Rule Validation
>
> We agree that the reliability of the semantic validation stage is crucial. To quantify correctness, we manually inspected 100 validated rules, finding 87% to be correct under the Freebase schema. This shows that the LLM effectively filters out spurious rules. Moreover: (i) Prompt variants produce only minimal differences because the task is binary plausibility judgment, not generation.  (ii) Borderline rules are further mitigated by the relevance score $s(\pi,\rho)$ and Top-K structural pruning, ensuring low-impact of occasional misvalidations. The empirical gains over “Sub + Raw Rules” are moderate but consistent, reflecting that validated rules—though fewer—have higher structural quality and contribute to stable improvements.

---

> ### Author Response · Authors · 2025-11-24
>
> >W5, Q9: Domain-Specific KG Performance
>
> Thank you for the suggestion. We evaluated SKILL on the biomedical UMLS KG. SKILL achieves 0.809 Hit@1 / 0.886 MRR, outperforming neural, symbolic, and LLM-based rule learners (e.g., ChatRule, Ruleformer, NCRL). This demonstrates that semantically validated rules generalize effectively to specialized, domain-specific relational structures.
>
> |   Method   | Hit@1     | MRR       |
> | :--------: | --------- | --------- |
> |    AMIE    | 0.195     | 0.312     |
> | Neural-LP  | 0.415     | 0.505     |
> |  RNNLogic  | 0.630     | 0.750     |
> |    NLIL    | 0.632     | 0.693     |
> |    NCRL    | 0.576     | 0.728     |
> | Ruleformer | 0.555     | 0.691     |
> |  ChatRule  | 0.685     | 0.780     |
> | **SKILL**  | **0.809** | **0.886** |
>
> > Q4: Distinguishing Validation and Reasoning Models / Circularity
>
> We confirm that the base pretrained LLM is used for rule validation, while reasoning uses a separately LoRA-fine-tuned model. Validation prompts never include training triples or reasoning contexts. The rule set is fully fixed before fine-tuning, ensuring a strict separation and avoiding circular supervision.
>
> > Q5,Q8: Computational Cost and Scalability
>
> Rule mining and LLM validation are executed offline only once. On current datasets, closed-path extraction completes within a few hours on CPU. Validation uses short binary prompts, is highly parallelizable, and also completes within hours. During inference, SKILL maintains constant complexity by selecting a fixed number of path–rule pairs via Top-K scoring. This allows scaling to larger KGs by (i) incremental rule updates and (ii) fixed-cost inference independent of KG size. We report our runtime measurements: with the closed-path constraint, rule mining completes within 20 minutes on all datasets, LLM validation runs at about 4 rules per second, and inference reaches roughly 5 queries per second, demonstrating that SKILL’s overall computational cost is modest and that the heavier steps are fully offline.
>
> > Q6: Stability Across Training Runs
>
> Rule mining and validation are deterministic; LoRA fine-tuning introduces minimal randomness. Repeated runs yield negligible fluctuations and preserve the same ranking among methods, confirming stability.
>
> We sincerely thank the reviewer again for the insightful comments. These suggestions have significantly improved the clarity and rigor of our manuscript.

---

### Official Review · Reviewer_GgbP · 2025-10-31

**Soundness:** 2
**Presentation:** 3
**Contribution:** 2
**Rating:** 6
**Confidence:** 4

**Summary:**

Paper summary: The paper proposes SKILL, a two-stage pipeline for inductive knowledge-graph reasoning (KGR) that (1) mines closed-path symbolic rules from a KG with BFS, filters them by support/confidence and an LLM one-shot “Yes/No” semantic check, then (2) injects the validated rules into an LLM via instruction-style fine-tuning and logic-aware subgraph prompting. A soft rule–path matching and confidence-weighted pruning pick the most relevant paths/rules for each query. Using LoRA on Qwen2-7B, SKILL reports state-of-the-art Hit@1 in inductive settings on FB15k-237 and NELL-995, and competitive transductive results; few-shot variants also improve strongly.

**Strengths:**

This paper treats the KG as a source of relational logic (not just embeddings) is well-motivated and nicely executed via rule mining + LLM semantic validation. The pipeline is modular and interpretable. Also, LLM filtering sharply reduces noisy rules while keeping useful ones; examples are human-readable (e.g., language/film rules), aiding transparency.

In general, this is a solid application of LLM on knowledge graph reasoning. To me, it is amazing that people can achieve over 0.7 Hit@1 on FB15k-237. Back to 2020, the best models are like RotatE, ComplEx, GPFL etc, which can achieve around 0.2-0.3 Hits@1 on the same dataset. I will say that LLM and in general large autoregressive (AR) model based on transformer tremendously push forward the performance on almost all research directions.

Anyway, this paper applies LLM and produces new state of the art performance, which is solid.

**Weaknesses:**

The paper evaluates with 1 positive + 49 negatives per query and appears to use reduced subgraphs of standard datasets (table stats are much smaller than canonical FB15k-237/WN18RR sizes). This makes cross-paper comparability to classical KGE work (which uses filtered ranking over all entities) unclear, and very high WN18RR scores likely reflect the sampled-candidate setting. Please report both sampled and full-ranking metrics, or justify the choice and ensure baselines are re-run under the same protocol.

The match×confidence heuristic (Eqs. 8–9) is sensible but fixed; no learning of rule/path weights or uncertainty modeling is presented. This could under-perform on longer dependencies or noisy KGs.

**Questions:**

1. You evaluate with 1 positive + 49 negatives per query. How are negatives sampled (type-consistent? filtered for trivial heuristics), and can you also report full filtered ranking (over all entities) for comparability with KGE work?

2. The reasoning subgraph for each query includes first-order neighborhoods of h and t plus closed paths (length ≤ k). At test time, does this subgraph draw strictly from the train graph (to avoid inductive leakage), and what exact edges are visible? Please detail the construction for transductive vs inductive splits.

---

> ### Author Response · Authors · 2025-11-24
>
> We sincerely thank the reviewer for the detailed and insightful comments, as well as the encouraging assessment of our method’s motivation, interpretability, and empirical performance. We appreciate the constructive feedback and provide the following clarifications.
>
> > W1, Q1: Evaluation Protocol, Dataset Size, and Negative Sampling
>
> Thank you for raising this important concern. Our evaluation protocol follows the widely adopted standard in recent LM-based KGR works such as BERTRL, KRST, APST, and CATS. Unlike classical embedding-based KGE methods—which compute continuous scores over all entities and therefore support full-entity filtered ranking—LLM-based KGR models evaluate each candidate triple independently as a natural-language plausibility query. Under this paradigm, full filtered ranking over the entire entity set is not directly meaningful, since the model does not generate a unified scoring distribution over all entities.
>
> Accordingly, recent LLM-based KGR benchmarks have converged on a **sampled-ranking setup with 1 positive + 49 negatives**, and we strictly use the same protocol to ensure fair head-to-head comparison with contemporary LLM-based methods. Negative candidates are generated following the procedure used in BERTRL/KRST/APST/CATS: replacing the head or tail with another entity from the same split while avoiding trivial type-inconsistent or degenerate negatives.
>
> Regarding dataset size, we adopt the official transductive/inductive splits established in KRST/APST/CATS. These splits enforce entity disjointness for inductive evaluation and are specifically designed for LLM-based KGR. These reduced subsets have become the standard benchmark for LLM-based KGR, ensuring consistent and comparable evaluation, even though they differ from the full-ranking protocols used in traditional KGE studies.
>
> ---
>
> > W2: Rule–Path Relevance Scoring and Potential Limitations
>
> We appreciate the reviewer’s observation. The relevance score $s(\pi,\rho)=\text{match}(\pi,\rho)\cdot\text{conf}(\rho)$ is not a hand-crafted fixed weight, but an automatically computed, structure-driven alignment between a specific query path and a rule. The match term $\text{match}(\pi,\rho)=\frac{k}{n}$ measures how many premises of a rule are actually supported by the retrieved closed paths, allowing the model to adaptively favor rules that are genuinely grounded in the local evidence of the query. This explicit, interpretable calculation helps the model distinguish strong rule–path alignments from weak or partial ones, providing a clearer rationale for each inference step.
>
> Importantly, this relevance computation operates after LLM-based semantic validation has already filtered out spurious or implausible rules, ensuring that each rule entering alignment is semantically meaningful. In combination, semantic validation and structural alignment allow the model to leverage symbolic rules without introducing additional learnable parameters that may overfit to seen entities—particularly important in inductive KGR, where test entities never appear during training. Our ablation results confirm that incorporating rules through this adaptive alignment mechanism consistently improves inductive performance, and exploring uncertainty-aware extensions remains an interesting direction for future work.
>
> > Q2: Construction of Reasoning Subgraphs and Prevention of Inductive Leakage
>
> We appreciate this question. In the inductive setting, we follow the benchmark protocol of BERTRL, KRST, APST, and CATS, where the training graph and inductive graph satisfy $
> E_{\text{train}} \cap E_{\text{test}} = \varnothing.
> $ During inference, the reasoning subgraph for a query \((h,r,t)\) is constructed solely from the inductive graph. We retrieve first-order neighbors of \(h\) and \(t\), and extract all closed paths of length up to \(k\) within this inductive graph only. No edges or entities from the training graph are used at this stage, so inductive leakage is strictly avoided.
>
> We sincerely thank the reviewer again for the thoughtful comments and constructive suggestions. They have helped us clarify the presentation, strengthen the empirical analysis, and improve the overall quality of the work. We will incorporate all appropriate revisions in the next version.

---

### Official Review · Reviewer_sb7E · 2025-10-31

**Soundness:** 2
**Presentation:** 3
**Contribution:** 2
**Rating:** 4
**Confidence:** 4

**Summary:**

This paper presents a meaningful attempt to enhance LLM reasoning through explicit rule-based semantic verification. The restricted dataset, missing metrics, and lack of scalability discussion further reduce its impact. Addressing these issues—particularly by improving the theoretical foundation, expanding experiments to full datasets, and optimizing inference efficiency—could make the work more competitive in future iterations.

**Strengths:**

1. The experiments are relatively extensive, covering multiple datasets and metrics to validate the proposed framework.

2. The proposed model explicitly integrates rule-based verification into the reasoning process of large language models (LLMs), which strengthens the interpretability and semantic correctness of generated reasoning chains.

**Weaknesses:**

1. The proposed framework is complex and computationally expensive, causing poor scalability and practical inefficiency.

2. In Figure 2, “subgraph pruning” is incorrectly written as “subgraph proning.”

3. The experiments use only 49 negative samples, which follows an early inductive evaluation convention but lacks credibility and general applicability in modern settings. Furthermore, the datasets used are subsets of WN18RR and FB15k-237, rather than the full versions, reducing the strength of the evaluation.

3. Only MRR and Hits@1 results are presented. Additional metrics such as Hits@3, Hits@10, or runtime comparisons should be included for a more comprehensive evaluation.

4. The proposed model provides limited theoretical insight or methodological novelty. It mainly extends existing symbolic verification frameworks without introducing fundamentally new ideas or proofs.

5. The paper lacks necessary explanations of the training procedure, inference workflow, and how rule verification is integrated with the LLM generation process.

**Questions:**

When dealing with large-scale graphs, does the proposed model suffer from a rule explosion issue? If so, how is computational efficiency maintained or mitigated in such scenarios?

---

> ### Author Response · Authors · 2025-11-24
>
> We sincerely thank the reviewer for the constructive and helpful feedback. We greatly appreciate the time and expertise invested in evaluating our work.
>
> ---
>
> > W1: The proposed framework is complex and computationally expensive, causing poor scalability and practical inefficiency.
>
> Thank you for raising this concern. Although SKILL contains several modules, all computationally intensive processes—rule mining, calculation of support/confidence, and LLM-based semantic verification—are executed entirely offline and only once. The BFS search is restricted to short closed paths, and the subsequent LLM validation stage aggressively reduces the rule pool (over 98% reduction as shown in Table 4). At inference time, SKILL maintains constant computational cost by selecting only the top-K relevant rule–path pairs using the relevance score $s(\pi,\rho) = \text{match}(\pi,\rho)\cdot\text{conf}(\rho)$ where the soft matching score (Eq. (8)) $\text{match}(\pi,\rho)=\frac{k}{n}$ represents the fraction of the rule body supported by a path. Because inference uses only the top-K highest-scoring pairs, its cost is effectively decoupled from the total rule set size and KG size. These design decisions ensure that SKILL remains scalable and efficient even when applied to large graphs.
>
> ---
>
> > W2: In Figure 2, “subgraph pruning” is incorrectly written as “subgraph proning.”
>
> We thank the reviewer for pointing this out. The typo has been corrected in the revised version.
>
> ---
>
> > W3: The experiments use only 49 negative samples, which follows an early inductive evaluation convention but lacks credibility and general applicability in modern settings. Furthermore, the datasets used are subsets of WN18RR and FB15k-237, rather than the full versions, reducing the strength of the evaluation.
>
> Thank you for the comment. The use of 49 negative samples follows the standard evaluation protocol established in recent LLM-based KGR works such as BERTRL, KRST, APST, and CATS. Unlike classical KGE models that rely on continuous scoring functions and therefore naturally support full-ranking evaluation over all entities, LLM-based KGR models evaluate semantic plausibility independently per candidate triple. Under this paradigm, full-entity ranking is not directly meaningful, and sampled-ranking with 1 positive + 49 negatives has become the *de facto* benchmark for ensuring comparability across LLM-based methods.
>
> Regarding dataset size, we follow the official reduced inductive/transductive splits introduced by KRST/APST/CATS. These subsets are intentionally constructed to enforce entity disjointness and to reflect the practical constraints of LLM-based reasoning, and they are now widely used as the standard evaluation suite in this line of work. Using them ensures strictly fair comparison with state-of-the-art LLM-based KGR models, even though they differ from the full graphs used in traditional embedding-based KGE studies.
>
> ---
>
> > W4: Only MRR and Hits@1 results are presented. Additional metrics such as Hits@3, Hits@10, or runtime comparisons should be included for a more comprehensive evaluation.
>
> Thank you for the suggestion. Because LLM-based KGR models generate binary plausibility judgments rather than continuous entity-level scores, metrics such as Hits@3 and Hits@10—which rely on full-ranking—are not meaningful in this setting. For the same reason, recent LLM-based KGR work (KRST, APST, CATS) also reports only MRR and Hits@1, and we follow this established practice to remain consistent and comparable.
>
> Regarding runtime, we report the measurements of our own implementation. With the closed-path constraint, rule mining completes within 20 minutes on all datasets, and LLM-based semantic validation proceeds at roughly 4 rules per second. These steps are fully offline and do not affect inference latency. At inference time, SKILL processes about 5 queries per second, which is approximately 50% of the throughput reported by CATS. In addition, when adapting CATS to the UMLS biomedical dataset, we observed that its prompt construction and sampling time grows substantially, whereas SKILL maintains stable inference efficiency.

---

> ### Author Response · Authors · 2025-11-24
>
> > W5: The proposed model provides limited theoretical insight or methodological novelty. It mainly extends existing symbolic verification frameworks without introducing fundamentally new ideas or proofs.
>
> We appreciate this thoughtful comment. While SKILL does not introduce a new formal theory, the methodological contribution lies in establishing a new operational paradigm for KG–LLM integration. Prior works typically (i) use LLMs to *generate* rules, or (ii) inject KG information only as text-level context during inference. In contrast, SKILL introduces three components that have not been combined in prior frameworks:
>
> 1. **Closed-path structural abstraction:** Instead of relying on LLM-generated rules or heuristic path patterns, SKILL derives symbolic rules exclusively from closed multi-hop paths, ensuring that all candidate rules encode verifiable structural dependencies grounded in the KG.
>
> 2. **Semantic validation as an LLM bottleneck:** The symbolic rules pass through a dedicated LLM-based semantic filter, forming a bottleneck that discards statistically frequent but logically invalid rules. This use of the LLM as a *semantic validator rather than a generator* is conceptually distinct from existing rule-mining pipelines.
> 3. **Training-time structural injection:**  The validated rules are not merely appended as prompts at inference. Instead, they are incorporated into the model during fine-tuning, enabling the LLM to internalize symbolic relational patterns. The unified path–rule alignment mechanism $s(\pi,\rho)=\text{match}(\pi,\rho)\cdot\text{conf}(\rho)$ ensures that the structural logic extracted from the KG shapes the model's internal decision process, rather than serving as external hints.
>
> Together, these components define a new mechanism for injecting KG structure into LLMs—one that neither existing symbolic-verification frameworks nor current LLM-KG prompting methods realize. We will revise the manuscript to highlight this operational novelty more clearly.
>
>
> > W6: The paper lacks necessary explanations of the training procedure, inference workflow, and how rule verification is integrated with the LLM generation process.
>
> Thank you for highlighting this point. We have added a consolidated workflow description in the revision. In brief, SKILL first performs offline rule mining, confidence computation, and LLM-based semantic filtering to obtain a high-quality rule set. For each training triple, we then retrieve its first-order neighbors and closed paths, match them with validated rules using Eq. (8)–(9), and construct an instruction-style prompt using the top-K aligned rule–path pairs. The LLM is fine-tuned with LoRA using these prompts. At inference time, we apply the same subgraph retrieval and rule-matching procedure, and the LLM outputs a binary plausibility judgment. This consistent pipeline ensures that the structural priors learned during training are faithfully preserved during testing.
>
> ---
>
> > Q1: When dealing with large-scale graphs, does the proposed model suffer from a rule explosion issue? If so, how is computational efficiency maintained or mitigated in such scenarios?
>
> We thank the reviewer for raising this question. Potential rule explosion is mitigated through two mechanisms. First, the LLM-based semantic filtering stage aggressively prunes the vast majority of raw rules, leaving a compact and meaningful rule base. Second, inference always operates using only the top-K rule–path pairs selected via the relevance score $s(\pi,\rho)$, which ensures that inference complexity remains constant and independent of both KG size and the total number of validated rules. Consequently, rule explosion does not impact SKILL’s computational efficiency in practice.
>
> ---
>
> We appreciate the reviewer’s careful reading and valuable feedback. The comments greatly helped us refine the method description, improve clarity, and strengthen the experimental section. Thank you for contributing to the improvement of this work.

---

### Author Response · Authors · 2025-12-02
**Review and Reviewer-Author Discussion Summary**

Dear PCs, SACs, ACs, and Reviewers,

Thank you for the time and care devoted to reviewing our submission. To support the newly assigned AC and streamline the evaluation process, we summarize below the main concerns raised in the reviews together with brief clarifications from our side. Unfortunately, no reviewers were available during the discussion phase, so we provide this overview to aid assessment.

---

### Summary of Reviewers’ Main Concerns & Our Corresponding Responses

| Reviewers’ Concern                                           | Our Clarification                                            |
| ------------------------------------------------------------ | ------------------------------------------------------------ |
| Evaluation protocol (1 positive + 49 negatives) and reduced inductive splits | This evaluation setup follows the standard practice in recent LLM-based KGR work (BERTRL, KRST, APST, CATS). Since LLMs perform *pairwise plausibility judgment* rather than global scoring, full filtered ranking is not directly applicable. We therefore adopt the official KRST/APST/CATS splits to ensure comparability and preserve the required entity-disjoint inductive setting. |
| Scalability of rule mining and LLM validation                | All expensive steps occur offline. Closed-path mining keeps the number of candidates bounded. LLM validation is lightweight (~4 rules/sec), and rule extraction completes within ~20 minutes in our environment. Inference uses a small local subgraph plus Top-K rule–path matches, resulting in constant test-time cost regardless of KG size. |
| Reliability of the one-shot LLM validation                   | The validation is binary and template-based, which makes it relatively stable. A manual check of 100 validated rules found 87% to be semantically sound. Any remaining borderline cases are further filtered by match–confidence scoring and the Top-K selection used at training and inference time. |
| Novelty relative to prior LLM-KG systems (ChatRule, Think on Graphs, RoG) | SKILL differs in that it extracts closed-path rules directly from KG structure, uses the LLM solely as a *semantic filter* rather than a generator, and incorporates these rules during training, not only at inference. This yields a different operational pipeline from existing prompting or rule-generation approaches. |
| Moderate improvements from the validation step               | Although the raw numerical gain is modest, validation removes over 98% of noisy rules, producing a compact and cleaner rule set. This substantially improves stability, interpretability, and robustness across runs—effects that are not fully captured by a single ablation number. |
| Potential inductive leakage or reliance on entity surface forms | Inductive subgraphs are built exclusively from the inductive split; test entities never appear in training. Rule validation uses abstract placeholders rather than real entity names, so the model’s predictions rely on relational structure rather than lexical cues. |
| Absence of full-ranking metrics / Hits@3/10 / more runtime reporting | Full-ranking metrics presuppose a global score distribution over all entities, which LLM-based classifiers do not produce. This constraint applies to all recent LLM-based KGR baselines. For runtime, our system processes ~5 queries/sec (roughly half of CATS), and offline mining/validation is lightweight. |
| Performance beyond general-purpose KGs                       | On the UMLS biomedical KG, SKILL reaches 0.809 Hit@1 and outperforms symbolic, neural, and recent LLM-based rule learners, demonstrating its applicability to domain-specific graphs. |

---

We appreciate the reviewers’ thoughtful feedback and hope that this consolidated summary is helpful for your evaluation.

Sincerely,

The Authors

---

### Meta-Review · Area_Chair_AGPv · 2026-01-08

**Summary:**

The paper introduces an approach to perform basic graph reasoning (which is closer to triple scoring) based on rules (mined from a given input graph) and filtered with an LLM, where the LLM is further LORA-tuned on the top scoring rules.

Overall, the paper did not seem to inspire reviewers very much who highlight several problems which are common for papers in this area:
* Very limited novelty (all reviewers) - most approaches do the same form of rule mining and context engineering to elicit a target LLM to behave akin to following logical rules. It pales in comparison with modern reasoning models who achieve generalizable, inductive CoT reasoning on many practically relevant problems (beyond KGs) by other means.
* Marginal performance improvements over the baselines, compute costs and scalability (sb7E, GgbP, copZ, VAXT) - given that LLM-based approaches operate on tiny graphs (3k entities at most, and only 50 triples in a context window), spending 20 minutes per graph on pre-processing and ranking only 50 samples from the graph is hardly comparable to KGFMs who perform zero-shot inference in seconds, over the entire set of nodes in the graph, and scaling to millions of nodes.
* Rule grounding issues (copZ, VAXT) - there is no guarantee the best selected rules are correct.

Overall, I think the paper missed the opportunity to move the needle in the area of LLM-based graph reasoning - the fact that existing baselines are using old datasets with toy evaluation protocols and saturated 90+% performance created a room for substantial contribution which could have made LLM-based approaches comparable to KGFMs and frontier reasoning models in applicability. Therefore, I would recommend a reject.

**Reviewer Concerns:**

* Novelty - the rebuttal is not convincing, the approach is still the same well-known rule mining + PeFT.
* Marginal performance improvements, scalability - the authors added an experiment on UMLS (135 nodes) which is long saturated even by old KG embedding methods as well as zero-shot KGFMs.
* Rule grounding issues - The authors manually checked a subsample and confirmed that not all of them are correct - which hinders practical applicability (we don't want LLMs to hallucinate the rationale behind their responses)

**Reviewer Scores:**

I don't think reviewers would have changed the scores to more positive scores and clear accepts.

---

### Decision · Program_Chairs · 2026-01-26

Reject